# Understanding the self-assembly dynamics of A/T absent 'four-way DNA junctions with sticky ends' at altered physiological conditions through molecular dynamics simulations

Akanksha Singh[1], Ramesh Kumar Yadav[2], Ali Shati[3], Nitin Kumar Kamboj[4], Hesham Hasssan[5,6], Shiv Bharadwaj[7]*, Rashmi Rana[8]*, Umesh Yadava[1]*

1 Department of Physics, Deen Dayal Upadhyaya Gorakhpur University, Gorakhpur, India, 2 Department of Physics, B.R.D. Post Graduate College, Deoria, India, 3 Department of Biology, Faculty of Science, King Khaild University, Abha, Saudi Arabia, 4 School of Physical Sciences, DIT University, Dehradun, Uttarakhand, India, 5 Department of Pathology, College of Medicine, King Khaild University, Abha, Saudi Arabia, 6 Department of Pathology, Faculty of Medicine, Assiut University, Assiut, Egypt, 7 Department of Biotechnology, Institute of Biotechnology, College of Life and Applied Sciences, Yeungnam University, Gyeongsan, Gyeongbuk, Republic of Korea, 8 Department of Research, Sir Ganga Ram Hospital, New Delhi, India

* shiv@ynu.ac.kr (SB); Rashmi.rana@sgrh.com (RR); u_yadava@yahoo.com (UY)

## Abstract

Elucidation of structure and dynamics of alternative higher-order structures of DNA such as in branched form could be targeted for therapeutics designing. Herein, we are reporting the intrinsically dynamic and folds transitions of an unusual DNA junction with sequence d$(CGGCGGCCGC)_4$ which self-assembles into a four-way DNA junction form with sticky ends using long interval molecular simulations under various artificial physiological conditions. The original crystal structure coordinates (PDB ID: 3Q5C) for the selected DNA junction was considered for a total of 1.1 μs molecular dynamics simulation interval, including different temperature and pH, under OPLS-2005 force field using DESMOND suite. Following, post-dynamics structure parameters for the DNA junction were calculated and analyzed by comparison to the crystal structure. We show here that the self-assembly dynamics of DNA junction is mitigated by the temperature and pH sensitivities, and discloses peculiar structural properties as function of time. From this study it can be concluded on account of temperature sensitive and pH dependent behaviours, DNA junction periodic arrangements can willingly be synthesized and redeveloped for multiple uses like genetic biomarkers, DNA biosensor, DNA nanotechnology, DNA Zipper, *etc.* Furthermore, the pH dis-regulation behaviour may be used to trigger the functionality of DNA made drug–releasing nanomachines.

**Data Availability Statement:** All relevant data are within the paper and its Supporting information files.

**Funding:** The study was supported by the Deanship of Scientific Research(DSR), King Khalid University, Saudi Arabia through the grant no. (R. G.P. 1/290.43). The authors also acknowledge with thanks the Sir Ganga Ram Hospital, New Delhi, India for financial support. The funders had no role in study design, data collection and analysis, decision to publish, or preparation of the manuscript.

**Competing interests:** The authors have declared that no competing interests exist.

## 1. Introduction

Four ways DNA junction, also called Holliday junction, can be formed during meiotic and mitotic cycles of cell division. The DNA junction may receive one of a few adaptations relying upon buffer salt focuses and the succession of nucleic bases nearest to the intersection. Holliday junction is an example of homologous DNA recombination, named after the sub-atomic scientist Robin Holliday, who proposed its reality in 1964 [1]. The Holliday junctions are vital in numerous kinds of hereditary recombination just as in two-fold strands break/repair. These intersections typically have a symmetrical sequence and are accordingly portable implying that the four individual arms may slide through the intersection in a particular example that safeguards base matching. Furthermore, four arm intersections like Holliday intersections show up in some practical RNA atoms [2]. Fixed Holliday junctions with asymmetrical sequence that lock the strands in a particular position were artificially made by researchers to examine their structure as a model for normal Holliday intersections [3].

Holliday junctions may exist in distinct types of conformational isomers with various coaxial stacking between the four two-fold helical arms, where Coaxial stacking is the inclination of nucleic acid blunt ends to tie one another by associations between the uncovered bases. The three possible conformers are: an unstacked structure and two stacked structures. The unstacked structure is an almost square planar broadened compliance and overwhelms without divalent cations, for example magnesium ion ($Mg^{2+}$) [4] because of electrostatic repulsion counteract and predominate the stacked structures. On the other hand, the stacked structures have two constant two-fold helical areas isolated by an angle of about 60º in a right-handed direction, where two of the four strands remain helical staying inside every one of the two two-fold helical spaces while the other two cross between the two areas in an anti-parallel fashion. The two possible stacked structures differ in which sets of the arms are stacked with one another and the rules of which are profoundly dependent on the base sequences closest to the junction. A few sequences bring about equilibrium between the two conformers while others highly favour a single conformer [4]. Moreover, cruciform structures including Holliday junctions can emerge to remove helical strain in symmetrical sequences in DNA supercoils [5] while four arm junctions likewise show up in functional RNA molecules, for example U1 spliceosomal RNA and the hairpin ribozyme [6] of the tobacco ring-spot virus. The study of the four-way junctions, therefore, is crucial in understanding the central dogma of the molecular biology. In addition to its biological significance, a four-way junction (Holliday junction) may be applied as an essential motif in DNA based nanotechnology applications, such as site Dye-DNA Holliday junctions aggregates applications in organic photovoltaics, non-linear optics, and quantum information systems [7].

The crystallographic or NMR investigation of Holliday junction helps in understanding and explaining the sequences and components influencing the inherent structures. Moreover, DNA molecules are very sensitive to numerous cellular environmental conditions, e.g., temperature, pressure, pH, alkalinity, salt concentration etc. [8]. For instance, an association for between the concentrations of ion DNA Holliday junction from the open to the stacked conformation has been determined [9, 10]. Moreover, significant role of temperature in various phenomena dependent on the DNA junction is also reported, such as DNA mismatched base pairs [11], DNA hybridization [12] and DNA nanotechnology [13]. Moreover, the effects of acidic and alkaline solvents or environment have been found to impart enormous significant information on the biochemical fluctuations in the DNA functionality and stability [14, 15]. The crystal structure of the Holliday junction which have been taken for the study comprises of the sequence d(CGGCGGCCGC)$_4$ [16], possess the potential application in nanotechnology. It has a rhombus structure with twenty base-pairs which form large cavities in the crystal

lattice [16]. Its motif consists of double helical arms whose strands end in a single DNA, commonly called as sticky ends [17]. Thus, the explained dynamical behaviour, including time dependent structural and flexibility data on DNA, can only be explained by their molecular dynamic simulations [18, 19].

Molecular dynamics is a tool complementary to experimental methods by which we can make our assumptions beyond the experimental limits and theoretically explain the effect of variations in different parameters [20–24]. Molecular modeling and simulations on Holliday junctions began with the elucidation of the stereochemistry by von Kitzing et al. [25, 26]. Later, several researchers used the molecular dynamics simulation methods in combinations with improved version of force fields to study the dynamics behaviour or induced conformational changes under alter physiological conditions. For instance, Yu et al. [4] reported a 6 ns MD simulation on the ApC junction based on the CHARMM27 force field [27], used the results to deduce conformational models for transitions within Holliday junctions, proposed a schematic framework for the classification of these changes, and provided theoretical evidence for a tetrahedral OPN junction. Simmons et al. successfully used MD simulations to explore ion binding sites in isolated Holliday junctions [28]. Adendorff et al. utilized MD to study the structure and dynamics of four-arm isolated Holliday junctions to understand the impact of base sequences on the junction structure [29] However, the behaviour of stick end Holliday junction d(CGGCGGCCGC)$_4$ under altered physiological conditions that essentially affect the symmetry and resultant dynamics or adopted conformational stability are not fully understood.

Thereof, the present study is an attempt to explain the thermodynamic stability and conformational changes of special type of Holliday junction d(CGGCGGCCGC)$_4$ with sticks ends under altered physiological condition, i.e., temperature, and pH values, in comparison to computed parameters with the known experimental results. Particularly, the present study aimed to find the atomic level interactions formed in the stick end Holliday junction d(CGGCGGCCGC)$_4$ in response to a particular temperature coupled with altered pH values. Given the significance of change in structural conformations for DNA-based nanotechnology applications, we conduct molecular dynamics simulations of DNA using an all-atom model in an explicit saltwater environment to reveal the mechanism of DNA denaturation. This study would allow to decipher the route to use of stick end Holliday junction d(CGGCGGCCGC)$_4$ for nanotechnology application, such as such drug packing and site targeted drug delivery.

## 2. Methods

### 2.1. Explicit molecular dynamics simulation

The crystallography coordinates of the DNA Holliday junction (HJ) with sticky ends for the sequence d(CGGCGGCCGC)$_4$ were taken from the PDB entry 3Q5C (www.rcsb.org) [16]. Initially, all atoms molecular dynamics (MD) simulation [30–32] for 100 ns interval was considered for the selected DNA HJ model at three different temperatures (200, 300 and 310K) by utilizing Desmond package [33]. Herein, the DNA HJ model was protonated at neutral pH (7.0 ± 2) and placed in Monte-Carlo equilibrated explicit transferable intermolecular potential (TIP3P) solvent using 10 Å × 10 Å × 10 Å simulation grid box. The complete system was then charge neutralized by suitable number of counter sodium ions under physiological salt conditions (0.15 M NaCl) using the system building module of Desmond package. The simulation box contains 14568 water molecules, 36 Na$^+$ counter ions and 1256 atoms of the junction molecule. The whole system was first energy minimized by the steps of steepest decent method to achieve a slope threshold of 25 kcal/mol-Å followed by conjugate-gradient energy

minimization method with an acceptable intermingling of root mean square (RMS) gradient of 0.1 kcal/mol-Å. The speeds were assigned at 100K to all the atoms agreeing to Boltzmann's distribution law and the temperature of the system was then raised from 100K to 200/300/310 K over 2.0/3.0 ps by scaling speeds as indicated by Berendsen calculation under standard pressure [22, 23, 34]. After equilibration, the system was subjected to MD trajectory generation as function of 100 ns under NPT ensemble conjugated with OPLS (Optimized Potentials for Liquid Simulations)-2005 force field [35] as implemented within the Desmond suite [33]. Herein, for the assessment of long-range Coulombic forces and Particle Mesh Ewald strategy was applied [36]. Also, the Verlet leapfrog algorithm [37] was utilized in the mathematical incorporation with a 1.0-fs time step length for minimization and 2.0 fs for dynamics. Additionally, hydrogen-particle bonds were compelled to their 'ideal' lengths. The constant Temperature and pressure were maintained through the Berendsen algorithms [34] during the MD simulation interval. The best simulated structure obtained at respective temperatures were considered for further investigation under altered physiological pH values, i.e., neutral (pH = 7) to acidic (pH = 5 and 6) and alkaline (pH = 8 and 9). Following the simulation trajectories were analysed for statistical mechanics, i.e., root mean square deviation (RMSD), root mean square fluctuation (RMSF), and radius of gyration (Rg), for each chain in the DNA HJ model using simulation event analysis implemented in the Desmond package [33].

## 2.2. Principal component analysis

Under statistical mechanics, ensemble of junction coordinates generated during MD simulation can be processed for Principal components to collect the essential motions or modes, which represents the total fluctuations contributing to observed variance in the system [38, 39], Thus, principal component (PCs) analysis was conducted on each simulation trajectory to discover the nucleotide displacements in terms of essential nodes under the influence of selected temperature and pH in the DNA HJ model. Initially, all the poses of the DNA HJ (5000 frames) were superimposed on the first frame of the trajectory, followed by computation of mean coordinates and the setting of these frames as the new reference set. Once all the frames aligned to the average structure, the covariance matrix was constructed, and phosphorous (P) atoms were considered to scrutinize the DNA motion using Bio3D package under R environment (https://www.r-project.org/) [40].

## 2.3. Conformational analysis of DNA junction

The backbone parameters and helicoidal parameters were determined using X3DNA [28]. In our study, 10 snapshots were extracted at every 10 ns interval from 100 ns dynamics by sampling every 20 ps. The following parameters were analyzed to describe the DNA structural deformability, including axis bend angles, slides, roll angles, twist angles, rises and groove widths. For analysis, reconstruction and visualization of three-dimensional nucleic acid structure, we have used the software X3DNA [41]. The X3DNA can handle antiparallel and parallel double helixes, single stranded structures, triplexes, quadruplexes, and other complex tertiary motifs found both in DNA and RNA structures. The analysis routines identify and categorise all basic interactions and classify the double helical character of appropriate base pair steps. The program makes use of a reference frame for the description of nucleic acid base pair geometry and a rigorous matrix-based scheme to calculate local conformational parameters and rebuild the structure from these parameters. Helocoidal parameters and torsion angle parameters are crucial in deciphering the folding and rotation of bases during dynamical pathway.

## 3. Results and discussion

### 3.1. Dynamic behaviour of DNA Holliday junction at altered temperature

**3.1.1. Conformational variation and parameters analysis at altered temperature.** The crystal structure of the sticky ended DNA HJ model (0.0 ns structures in Fig 1a–1c) consist of the four strands associated to form a right-handed, antiparallel four-way junction with Watson-Crick base pairing. The arms of the DNA junction stack on each other in pairs and form the stacked-X conformation as in other Holliday junction crystal structures [42]. Each arm consists of 4 base pairs d(CCGC) and d(GCGG) while the sticky ends are observed due to two unpaired bases d(CG) at the 5' ends. The examination of each strand exhibits the absence of two-fold axis through the centre of the DNA junction [16]. As it has been discussed by Venkadesh *et al.*, [16], the $J_{twist}$ and $J_{roll}$ values for the sequence d(CGGCGGCCGC)$_4$ are respectively larger and smaller than in other reported sequences, indicating the easy accessibility of minor grooves at the crossover. At first, full-atomistic MD simulations were conducted for the selected DNA junction to assess the thermodynamics stability in solution under three different temperatures (200, 300, and 310K) and conformational properties of the simulated models were compared to that of crystal structure.

The simulation trajectory for the DNA HJ model at 200K, and pH 7, showed some variations in the snapshots taken at the periodic interval of 10.0 ns (Fig 1a). A close observation of the snapshots indicates more stretching at 30.0 ns and this trend continues throughout the dynamics, making fluctuations more visible, when we observe the snapshots at 80.0 ns, 90.0 ns and 100 ns. It appears that the middle strands have got stretched from the 5' end towards the 3' end. Although no such stretching on the upper part of the strands is observed at the junction point. The presence of three hydrogen bonds in G:C base pairs is probably keeping the middle portion of the first strand intact along the helix axis thereby maintaining the planarity vis-à-vis the G:C base pairs of the middle strand, though, the structural integrity is being maintained by the junction during the entire course of molecular dynamics simulation. Again, the molecular dynamics simulation was repeated at 300K by taking the initial structure and a series of snapshots were extracted at periodic intervals of 10.0 ns for conformational variation analysis (Fig 1b). A close observation reveals that when the temperature is raised to 300K, in the neutral condition, the junction fails to maintain the structural integrity and finally the strands separate larger. Fig 1b reveals that the structure of the junction remains stable from 0.0 ns to 80.0ns with a little change in orientation of the base pair plane. But as the time raised to 90.0ns there is an abrupt disturbance in the conformation of the junction and the junction get breaks. But again, at the simulation interval of 100.0 ns, the junction tried to regain its initial configuration with changed conformation, suggested as consequences of thermal vibrations which contribute to the instability of the DNA junction.

The snapshots taken at periodic intervals of 10.0 ns during MD simulation at 310K are shown in Fig 1c, reveals variation in DNA junction especially at the central region of the junction model during the 100 ns MD simulation interval. Analysis of the extracted snapshot demonstrates the flexibility in the base pairs located mostly at the middle portion of the junction. At this temperature, none of the snap shots exhibit abrupt disturbance that indicate that the abrupt fluctuations are momentarily. Also, bending at the junction is observed, contributed by the conformation change in the base pair plane with respect to adjacent bases getting closure to each other. The snapshot at 30.0 ns seems to indicate that the junction is regaining its initial structural conformation. Only after 50.0 ns snapshot onward, a clear indication of folding of the DNA junction model is observed and the clear changes in the model becoming obvious at the snapshots taken at 80.0, 90.0, and 100.0 ns respectively.

Furthermore, the individual backbone torsion angles (α, β, γ, δ, ε, ζ, η, and χ) for all the forty nucleotides in the DNA HJ junction model were computed and represented in

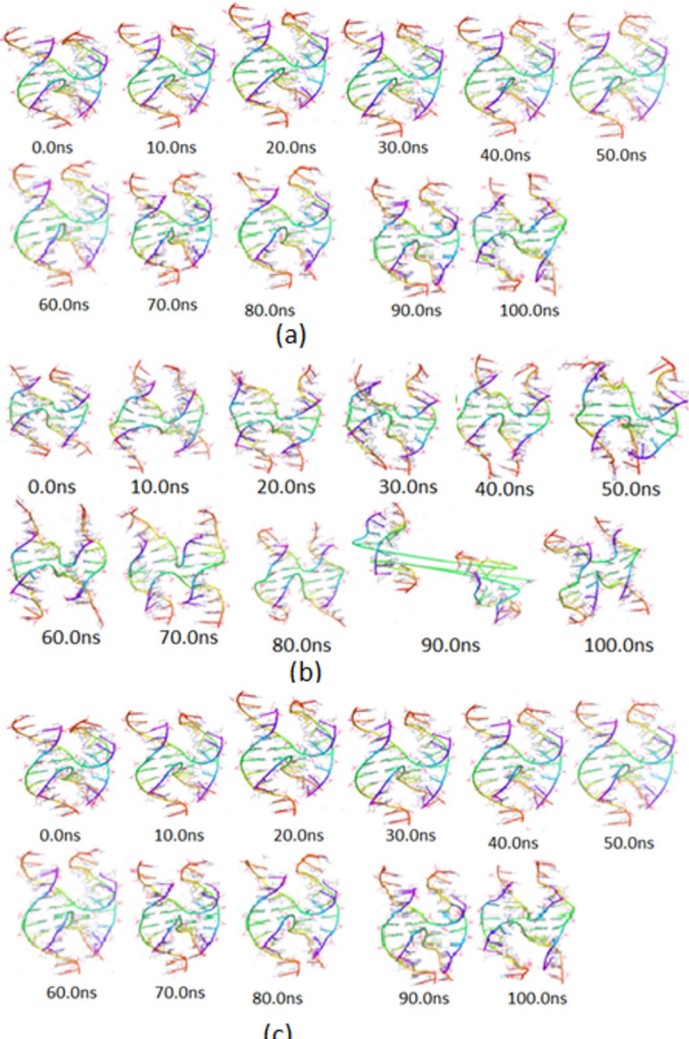

**Fig 1. A series of snapshots taken at periodic intervals of 10.0 ns during MD simulation of the four–way DNA junction at neutral pH and temperature (a) 200K, (b) 300K and (c) 310K, respectively.**

supporting information S1a–S1d Table. S1a Table is shown for the initial structure, i.e., pre-dynamics. While S1b Table depicts the torsion angles calculated for MD simulation done at 200K temperature for 100 ns time. S1c Table represents the torsion angles calculated for MD simulation done at 300K for the time of 100 ns, while S1d Table demonstrate the torsion angle parameters at 310K. The term ±Gauche (±g) and trans (t) have been used to refer the broad conformational space and Klyne-prelog notation such as *syn*, anti, synclinal(*sc*), anticlinal (±*ac*) have been used for more detailed analysis [43].

From S1a Table, we have observed that there is a considerable correlated conformational activity between torsion angles α and β, these torsion angles oscillate within gauche(*g*-) and trans(*t*) regions respectively for most of the residues. There is symmetric absence of torsion angles α and β for residues 1, 11, 21 and 31. Similar symmetric absence of torsion angles ε and δ are found for the residues 10, 20, 30 and 40, respectively. The torsion angle γ shows considerable dynamical excursion in the gauche (*g*-) regions to gauche (*g*+) regions with exception of residues 6, 20, 31, and 40, which are in *trans* regions. The torsion angles δ, show anticlinal (*ac*)

regions for most of the residues with exception at residues 5, 14, 23, 24 and 25. They are in trans (*t*) regions. The torsion angles ε and ζ oscillate freely in–anticlinal (*-ac*) and -synclinal (*-sc*) regions respectively. Similar trends are observed for glycosyl angle (χ), it hovers throughout the trajectory in–*sc* to–*ac* regions for all the residues of the molecule.

Even though the sugar rings show considerable dynamical excursion into the C1'-endo to C4'-endo regions as shown in S1a Table. We observe that most of the parameters are in proximity with the initial conformational parameters. It means that the structural integrity is maintained throughout the dynamics. However, when the temperature is raised to 200K and 300K the conformational parameters are altered and stability of the molecule decreases, the values are represented in S1b and S1c Table, respectively. On comparing different conformational parameters obtained by X3DNA program of the original structure with the standard values of corresponding conformational parameters of different types of DNA, we observe that different parameters of the original structure are in proximity with the expected ranges for B type DNA. For example, sugar conformational parameters *viz* puckering are C1′—endo, C2′-endo, C1′-exo, C3′-exo and C4′-exo and C3′-endo which is like puckering values of B-DNA. Similarly on observing the local base pair and step parameters of the original structure like buckle value is up to 20º (+*ve* or–*ve*) which is also like the buckle value of B-DNA [44, 45].

**3.1.2. Helicoidal parameters analysis.** The helicoidal parameters for the initial structure and the structures obtained after 100 ns molecular dynamics simulation, in neutral pH, at three different temperatures viz; 200K, 300K and 310K are tabulated in Supporting Information S2a–S2d Table, respectively. All these parameters have been calculated using X3DNA software [41]. The subset of the helicoidal parameters, i.e., Shear, Stretch, Stagger are the distance parameters; and hence, measured in angstrom (Å) unit. A close observation of these values, presented in the Tables, indicates that these parameters are relatively stable and shows the average values in the vicinity of zero. All these displacements are either zero or are very close to zero for both A and B form of DNA [45]. Any variation from zero effects the stacking of the base pairs and results in the instability of the complex. The other subset parameters Buckle, Propeller and opening are angle parameters and hence measured in degrees (º). Buckle values in the Tables shows oscillating values with extreme adjacent to the hinge points. The junction shows negative values of propeller for all the base pairs in the S2a Table. Similar trends is followed in the S2b–S2d Table with exceptions at 6, 10 and 12 numbered base pairs in the S2b Table and 9, 10, 12 base pairs in S2c Table having positive values.

The angle parameter ′opening′ shows consistent positive and negative values as usual. The negative values of opening are an indication of the opening towards the minor groove side. In present case no specificity has been observed for the opening of the base pairs. The parameters Shift, Slide and Rise are distance parameters and are also measured in angstrom (Å) unit. Here, we observe these values are near B- form DNA, showing oscillating values. The angle parameters––Tilt, Roll and Twist, are also plotted in Tables and the variation in these parameters at various steps are found for A and B forms DNA and any change in these parameters may be regarded as dislocations as a natural consequence of the finite backbone Chain length from one base pair to the next. Such correlations have also been observed in other B-DNA helices. The positive sense of Roll parameter indicates compression on the minor groove side. It means that the Roll and Tilt parameters define local expansion of the inter-base pair displacements with respect to minor groove side. The calculations described herein provide details on dynamical behaviour of the DNA junction model and the conformation at the initial conformation.

We observe from corresponding data that the parameters remain unaffected. This indicates that the junction is stable. Now on raising the temperature to 200K, it is observed that there is fluctuation in the values of parameters, this shows that the junction gets a little bit unstable at this temperature. On further raising the temperature to 300K, and 310K there is abrupt change

in the values of parameters, it shows that the junction gets completely unstable. This is because of the fact that kinetic energy of atoms increases with the rise of temperature that leads to the enhanced vibrations and instability which finally results into the disruption of the DNA junction.

**3.1.3. Temperature dependent RMSD analysis.**   To demonstrate global convergence in the DNA model, root mean square deviations (RMSD) was computed of the MD simulated structures at different temperatures (200, 300, and 310K) with respect to the initial structure as function of 100 ns interval (Fig 2a–2c). Analysis of RMSD for the DNA Junction at 200K showed considerable deviations (2 ± 0.10 Å) in all the four Chain (A-D) in the model (Fig 2). In totality, considering minor changes in torsion angles, and helicoidal structure, the small RMSD indicates that the DNA junction structure is considerably converged at 200 K. Convergence and stability profiles in Fig 4(a) indicates that during simulation the MD structure transits quickly to a stable equilibrium position, with reasonably small oscillations centred on ±0.70 Å for the combined Chains. From the RMSD plot shown in Fig 4(b), we observe that when the MD simulation was performed at 300K, it is seen that the RMSD of the heavy atoms, when the system is constrained, varies in a small range centred around 2 and 4.3 Å for all the four Chains (A, B, C & D), up to 60.0 ns dynamics. After 60.0 ns of dynamics, all the four strands become distinguishable and oscillation range extends from 5 to 6 Å. This indicates the increase in thermal fluctuations making the base pairs perturbed.

Finally, the RMSD shown in Fig 2c is the plot for the MD simulation performed at 310K. Here, we observe that for all the four Chains of the junction, the deviation which was initially centred around 2.0 ± 0.90 Å gets shifted to 4.0 ± 1.20 Å till end of the simulation. This may be due to the removal of constraints and contribution of entropy to achieve the stable conformation. A particularly high RMSD demonstrates the breaking of junction. Comparing the RMSD of the DNA junction structure as shown in Fig 2(a)–2(c) respectively, at three different temperatures, we conclude that DNA junction is temperature sensitive, any small amount of temperature variation alters the orientation of nucleic acid base pairs making the DNA a crucial molecule for further applications.

**3.1.4. Temperature dependent RMSF analysis.**   Root means square fluctuation (RMSF) measures the average deviation of the individual residues over time from a reference position. RMSF plot of the DNA junction structure during molecular dynamics simulation at different temperature, i.e., 200K, 300K and 310K, is depicted in Fig 3. We observe that during molecular dynamics simulation at 200K, the RMSF of all the four chains was < 2.5 Å and no substantial abrupt fluctuations were observed during the entire course of molecular dynamics of 100 ns. Likewise, RMSF analysis of chains at 300K depicted an increase in RMSF values, where highest RMSF of < 11 Å was noted for the Chain B against other three Chains A, C, and D (<8 Å). Whereas further increase in temperature at 310K showed similar pattern for the RMSF values of DNA junction chains as observed in 300K, except Chain C showed higher fluctuation (<10 Å). Notably, RMSF values for all the chains in DNA junction model treated at three different temperatures, only highly fluctuations were noted at the terminal residues. Together, as observed in the RMSD, a change in temperature to 310K (close to human body temperature) showed reduction in local rotation against 300K and higher than the 200K.

**3.1.5. Temperature dependent Rg analysis.**   The radii of gyration computed from the 100 ns MD simulation interval for all the four Chains in DNA junction model at selected temperatures (200, 300, and 310K) is depicted in Fig 4. When molecular dynamics simulation was performed at 200K temperature, the initial radius of gyration for Chain A was found to be 12.8 Å, for Chain B it was 12.8 Å, for Chain C it was 11.7 Å and for Chain D it was 12.2 Å. During dynamics, there is slight increase in the values for Chain A and Chain C at about 10.0 ns dynamics run and then start decreasing. Moreover, radius of gyration of Chain A and Chain C

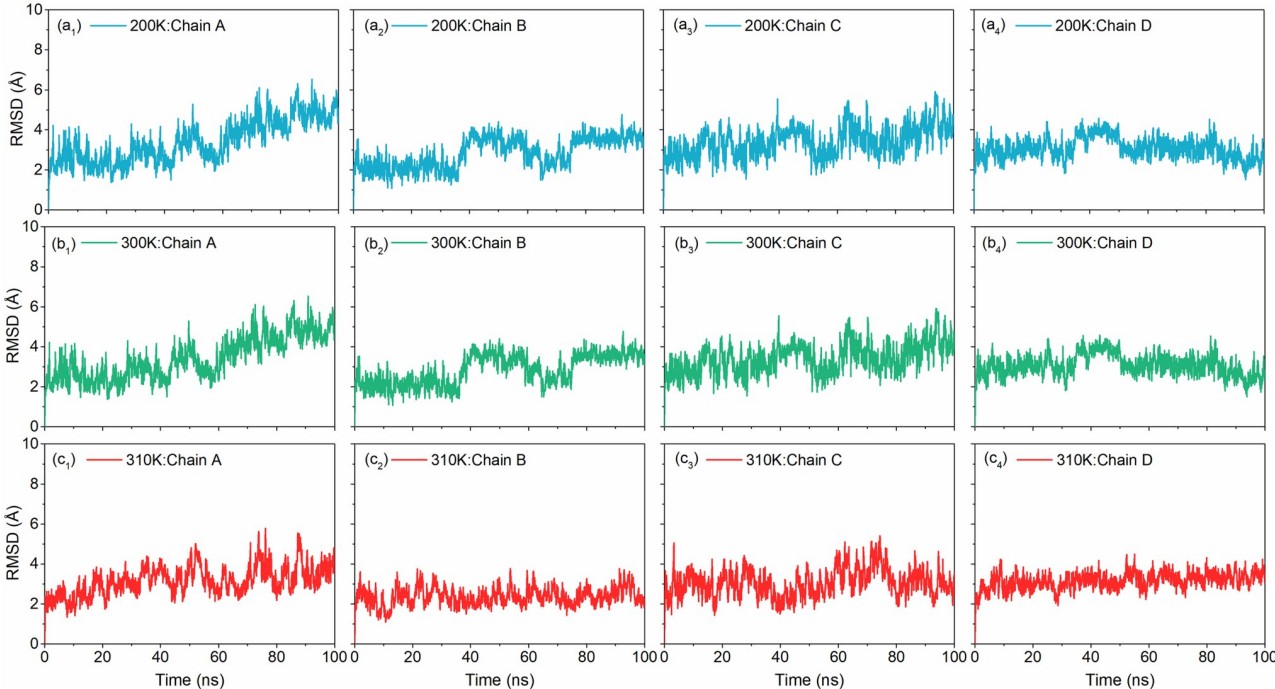

**Fig 2. RMSD plots of the MD simulations of the DNA junction at (a) 200K, (b) 300K and (c) 310K.**

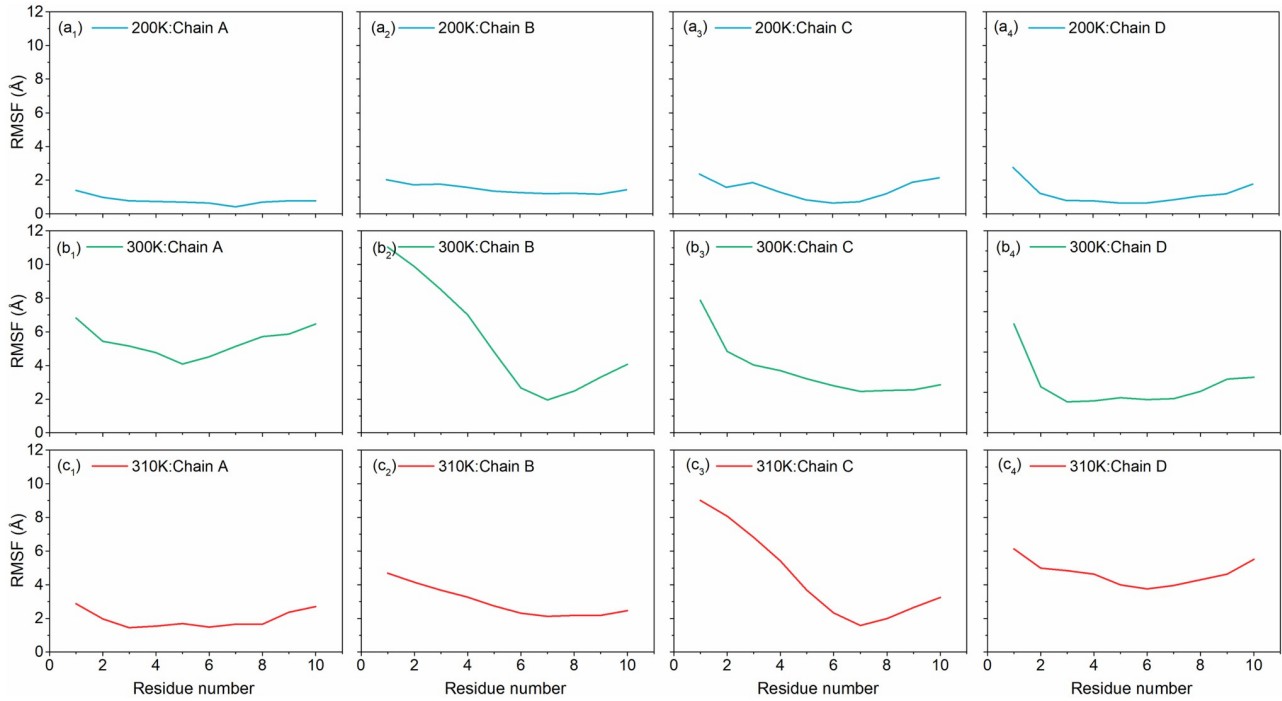

**Fig 3. RMSF plots of the MD simulated DNA Junctions at (a) 200K, (b) 300K, and (c) 310K respectively.**

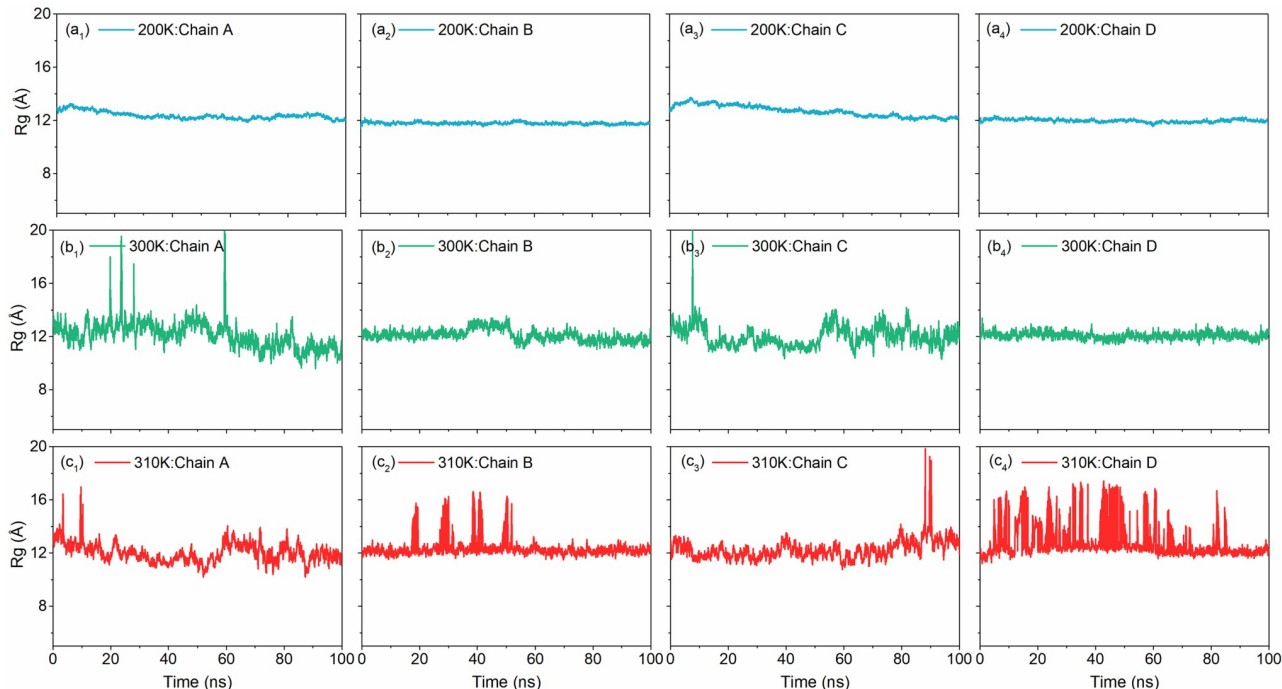

**Fig 4. Variation of radius of gyration (Rg) of all the strands of the DNA Junction with simulation time at(a) 200K, (b) 300K, and (c) 310K respectively.**

attains the highest values of 13.2 Å and 13.7 Å respectively. At the end of simulation, the radius of gyrations of Chains A and C decreases to 12.0 Å and 12.2 Å, respectively. The radius of gyrations for Chain B and Chain D maintain their initial values, with very small fluctuations, to 11.7 Å and 12.1 Å, respectively. For MD simulation performed at 300K, radius of gyration for all the four Chains A, B, C and D are around 12 Å for initial structure and 11 Å for final structure. Although some spikes in Chain A and Chain C were also observed indicating the instability of the junction at 300K. Similarly, all the chains in DNA junction showed occasional high peaks while maximum numbers of fluctuations in Rg values were noted for the Chain D during the 100 ns MD simulation at 310K. Besides, all the chains showed Rg value < 14.5 Å during the MD simulation performed at 310K.

**3.1.6. Essential dynamics of temperature dependent trajectories.** To understand the essential motions or displacements in the DNA model treated at different temperatures with neutral pH, essential dynamics in terms of principal components was performed on the generated complete simulation trajectories Herein, each trajectory was analysed in terms of screen plot, which represents the total variance in essential motion adopted by extracted principal component and the score plots, which indicate the projection of the data onto the span of the principal components (Fig 5). Initially, analysis of scree plot for the DNA junction simulated at 200, 300K, and 310K showed rapid reduction in motion within three PCs followed by a state of equilibrium. For instance, the PC1 showed 61.2, 69, 74.2% variance for 200, 300 and 310K in the respective dataset (Fig 5). Notably, most of the variance was covered by first three PCs, hence, these datasets were considered for further score plot analysis. The score plot analysis for the DNA trajectories computed at 200K showed compact clusters for the data by comparison to the dataset generated at 300 and 310K (Fig 5). These results support the finding from the conformational analysis and statistical mechanics analysis that lower temperature reduced the

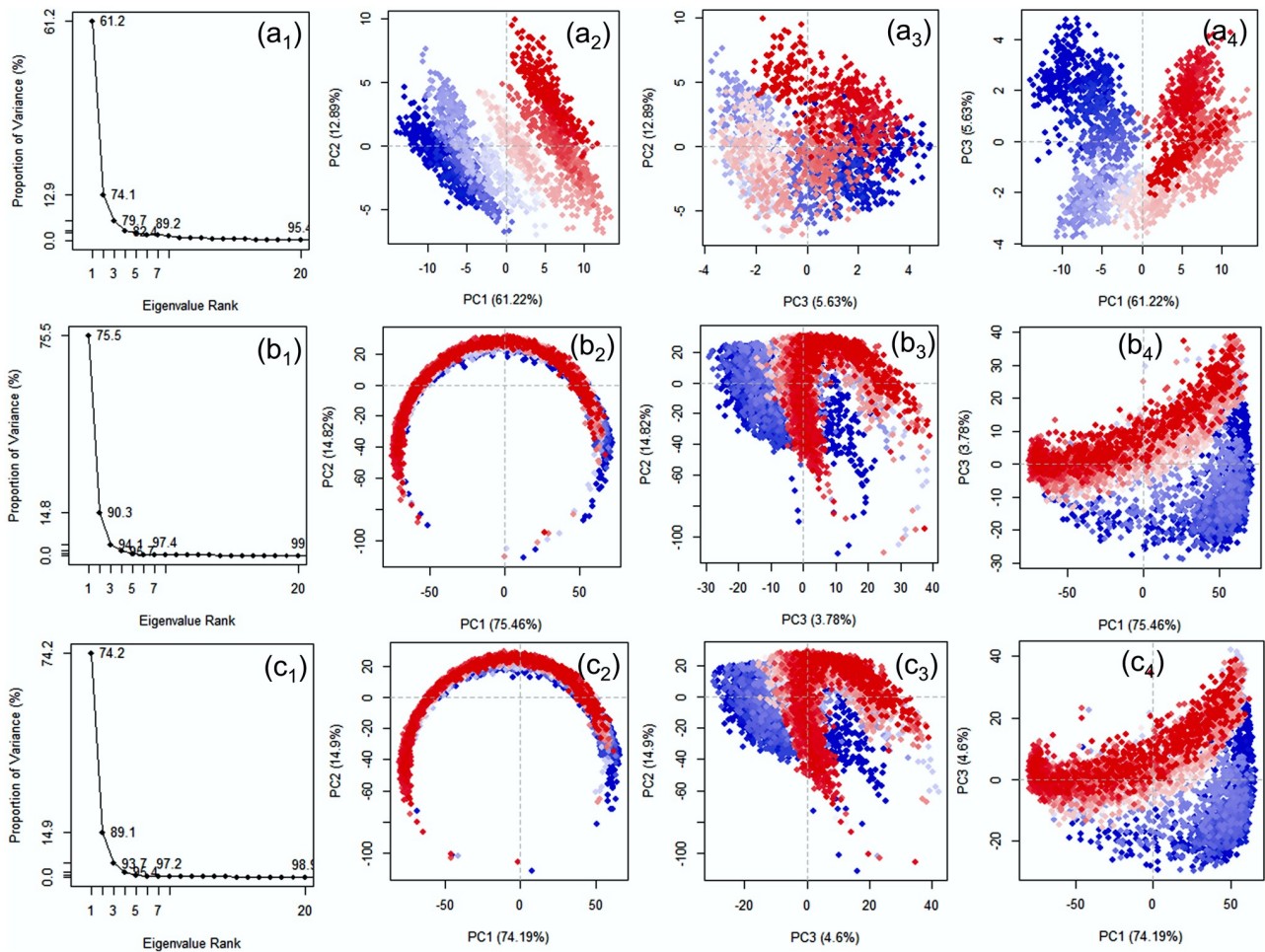

**Fig 5. Principal component analysis for the simulation trajectories at 200K (a₁-a₄), 300K (b₁-b₄), and 310K (c₁-c₄).** In the sub-figures (a₁, b₁, c₁), the X-axis (Eigenvalue Rank) represents the principal components (PCs) while Y-axis represents the percentage of total variance explained by each individual principal component. The scatter plots represent the distribution of extracted essential motion between two PCs while colour change from red to blue through white marks for the periodic fluctuations experienced by the residue during 100 ns MD simulation.

fluctuation in the DNA model. However, an increase in temperature, an increase in vibration for the DNA model was noted, suggested to contributed by the total entropy and enthalpy of the system. Of note, distinct clusters formations were also noted at 200 and 300K but overlapping clusters were observed at 310K. These observations suggested that DNA junction at 310K experienced least repulsive forces within the chains, which may further be attributed to the stable conformation under normal physiological condition.

### 3.2. Comparison of temperatures at altered pH on DNA Holliday junction

Based on the DNA junction simulation performed at three different temperatures, the considered model showed stability at 200K. However, DNA junction also exhibited considerable stability at 310K with comparison to 300K at neutral pH. Also, it has already been reported in literature that changes in pH could denature DNA [46] and plays a significant role for retaining the stability, molecular electronics, and other properties of the macromolecules [47, 48]. Extreme pH conditions lead to heightened structural and chemical alterations prominent to DNA damage that could have insightful biological concerns [49]. Therefore, we have carried

out the further molecular dynamic's simulation investigations on the junction at 300 and 310K by altering pH values ranging from 5 to 9. Herein, various conformational parameters and post simulation trajectory analysis, including RMSD, RMSF, Rg and essential dynamics, was studied and compared to that neutral pH value.

**3.2.1. Conformational variation and parameters analysis at constant temperatures coupled with altered pH values.** Torsion angle and helical parameters of the simulated structures at various pH values have been presented in S3 and S4 Tables. Torsion angles α and β oscillates between -gauche and trans(τ) regions for almost all the simulated structures. Similar trends have been observed for the other torsion angles too. The torsion angle δ, ζ, and χ exhibit small deviations from the B-form of DNA for both the strands of the nucleic acid duplexes. The base pair step parameters roll, twist and slide have the values close to those of B-form of the DNA structure (S5 and S6 Tables). The most variations have been observed only in the case of pH = 8, where the observed roll, twist and slide parameters are 3.13º, 32.2º and -0.16Å at the place of usual B-form values of 2º, 36º and 0Å, respectively.

**3.2.2. Minor and major groove width analysis.** The minor and major groove widths, obtained through X3DNA, demonstrate the slight departure of groove widths at different pH values (Table 1). The refined average minor groove widths at the 4th (GC/GC) and 12th (GC/GC) steps are 12.5 Å and 12.7 Å respectively at the neutral pH. However, the minor groove width has been reduced at 4th step to 8.6Å for pH = 5, 6 & 9 and increased to 13.6 Å at pH8. Similarly, the reduced refined minor groove widths at 12th step are found to be 10.1 Å for pH = 5 & 9, 12.9Å for pH6 and 11.8Å for pH8 respectively. Except at pH8, major groove widths do not show several variations instead they are within the range 15.9 to 16.6Å at 4th step while 16.7 to 19.5Å at 12th step. For the pH8, these major groove widths are 18.6 Å and 21.1 Å respectively.

The phosphorous displacement parameter Zp also provides the significant information about the forms of the DNA structure. Zp < 0.5Å indicate the nucleic acid duplex in B-form while Zp > 1.5Å usually represent A-form DNA structure [45, 51]. The calculation results demonstrate that most of the duplex structures at pH range 5–9 falls into the category of B-form structure. In a few cases the $Z_p$ values lies within 0.5–1.5 Å, but the values never go towards A or Z–form of DNA structures.

The width of minor and major groove widths at 300K and 310K at various pH values are shown in Fig 6. It can be observed at 300K temperature that the minor groove width at 4th step varies with pH variations and takes values ranging from 8.3 Å at pH 9 (80.0ns) to 15.0 Å at pH 5 (60.0ns). At pH 7 the variations observed are 8.9 to 12.1 Å. Minor groove width at 12th step has somewhat random variations within a small range except pH8. The longer minor groove width of 14.9 Å has been observed at 10ns dynamics pose of pH 8, while other larger minor groove widths observed are 14.6 Å at pH 7, 14.4 Å at pH 6 respectively. Apart from these, the minor groove widths obtained are within the range 9.9 Å to 13.7 Å at the 12th step of the DNA junction. Major groove widths at the 4th step have been obtained mostly within the range 14.7 Å to 22.9 Å. Other estimated widths outside of this range are 25.8 Å at pH5, 25.8 Å at pH7. The major groove width at pH 7 increases with the duration of dynamics and become maximum at 70.0 ns dynamics, after that, it starts decreasing and attains the value 21.9 Å at the end of the dynamics. Therefore, we can say that the junction become more unstable at pH 7. Similar trends are observed in the case of pH 8 and 9 while in the case of pH 5 and pH 6, the major groove width decreases with dynamics, indicating less instability in acidic medium. The average major groove width at 12th step shows increasing fluctuations with the dynamics. In beginning the major groove width lie within 17.0 to 17.7 Å which have been observed within the range 16.0 to 20.1 Å at the end of dynamics. The major groove widths at 12th step, have somewhat increased values, except at pH 7.

**Table 1. Minor and major groove widths (direct P-P distances) [50] at 100K and various pH values.**

| Step | Minor groove width (Å) | | | | | Major groove width(Å) | | | | |
|---|---|---|---|---|---|---|---|---|---|---|
| | pH | | | | | pH | | | | |
| | 5 | 6 | 7 | 8 | 9 | 5 | 6 | 7 | 8 | 9 |
| 3 GG/CC | 13.1 | 12.7 | 15.0 | 17.5 | 13.1 | 17.1 | 18.9 | 15.9 | 16.2 | 17.6 |
| 4 GC/GC | 8.6 | 8.6 | 12.3 | 14.1 | 8.6 | 16.6 | 15.9 | 16.1 | 19.0 | 16.6 |
| 5 CC/GG | 9.9 | 9.9 | 12.1 | 13.2 | 9.9 | 18.1 | 16.6 | 16.6 | 22.7 | 18.1 |
| - - | - - | - - | - - | - - | - - | - - | - - | - - | - - | - - |
| 11 GG/CC | 14.0 | 13.8 | 15.6 | 14.2 | 14.0 | 18.1 | 19.3 | 16.2 | 18.2 | 18.1 |
| 12 GC/GC | 10.1 | 10.9 | 12.7 | 11.9 | 10.1 | 19.5 | 19.8 | 16.7 | 21.1 | 19.5 |
| 13 CC/GG | 10.3 | 11.1 | 12.2 | 12.2 | 10.3 | 17.8 | 19.3 | 18.4 | 23.2 | 17.8 |

At 310K, the width of minor groove at the 4th step has less variations within the range 9.6 to 15.2 Å. Larger fluctuations have been observed in the beginning of dynamics in the case of pH 8 & 9 whereas groove widths at pH 7, 6 and 5 have less fluctuations during the entire course of dynamics. However, the minor groove widths have increased after the dynamics, designating increased instability of the DNA junction. Minor groove width at the 12th step, varies within the range 9.2 Å to 15.0 Å, larger being in the case of pH 6 while smaller values are observed at pH 8. In neutral case minor groove width variation is smaller as compared to variations for other pH values. The major groove widths at 4th step have smaller fluctuations and reduced at the end of dynamics, except the case of pH 8. Comparatively larger fluctuations have been

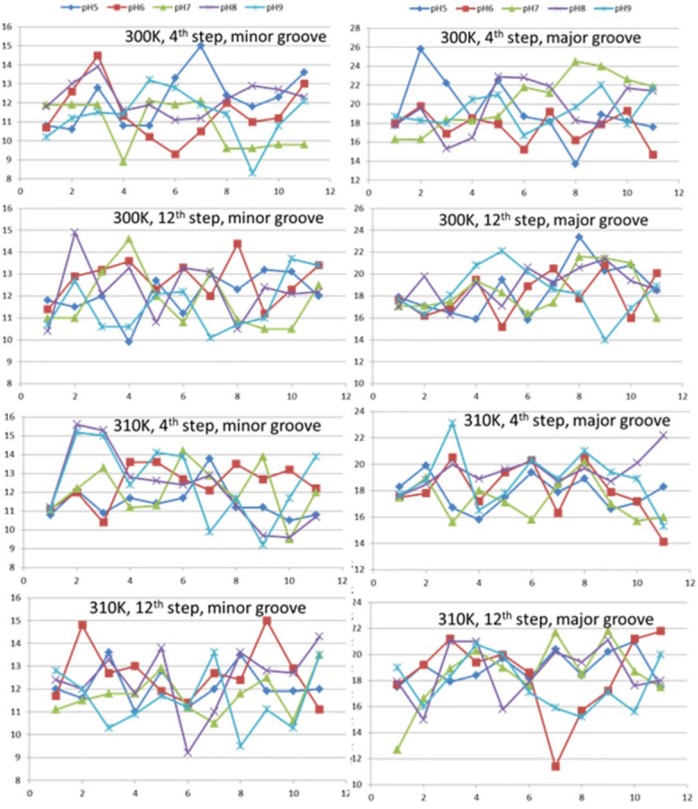

**Fig 6. Variation of average minor and major groove widths at 300K and 310K at altered pH values.**

observed in the case of pH 8 and pH 9. In other cases, the major groove widths have been estimated within the range 15.7 Å to 21.0 Å. At the 12th step the major groove widths have variations within the small range of 15.0 to 21.1 except the values at pH 6. Despite of these variations in minor and major groove widths, the nucleic acid structures are right-handed with B-type DNA at the helical regions.

**3.2.3. pH dependent RMSD analysis.** RMSD analysis was performed for the DNA junction simulation trajectories collected at two different temperatures and altered pH values. Interestingly, by comparison to pH7 as reference dataset, RMSD analysis of individual chains at 300 and 310K under acidic conditions (pH 5 and 6) showed deviations < 4.4 Å and < 5.6 Å, respectively high deviation ~ 6.1 Å was noted in the Chain C under 310 K at pH 5 around 60–80 ns (Figs 7 and 8). Likewise, under the alkaline conditions (pH 8 and 9), DNA junction

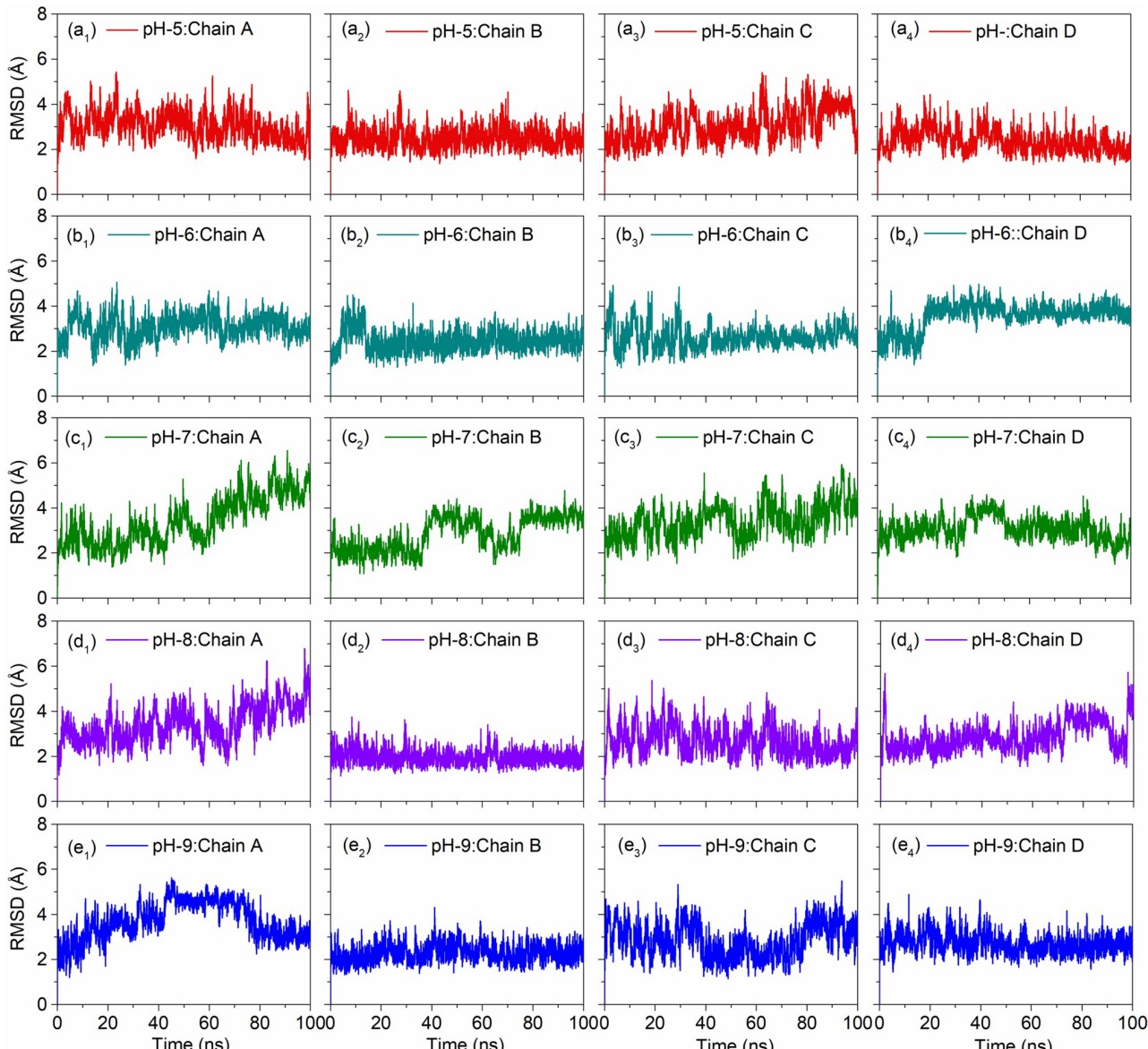

**Fig 7. RMSD plots of the MD simulations of the DNA junction under 300K with altered pH, viz.** ($a_1$-$a_4$) pH5, ($b_1$-$b_4$) pH6, ($c_1$-$c_4$) pH7 as reference, ($d_1$-$d_4$) pH8, and ($e_1$-$e_4$) pH9.

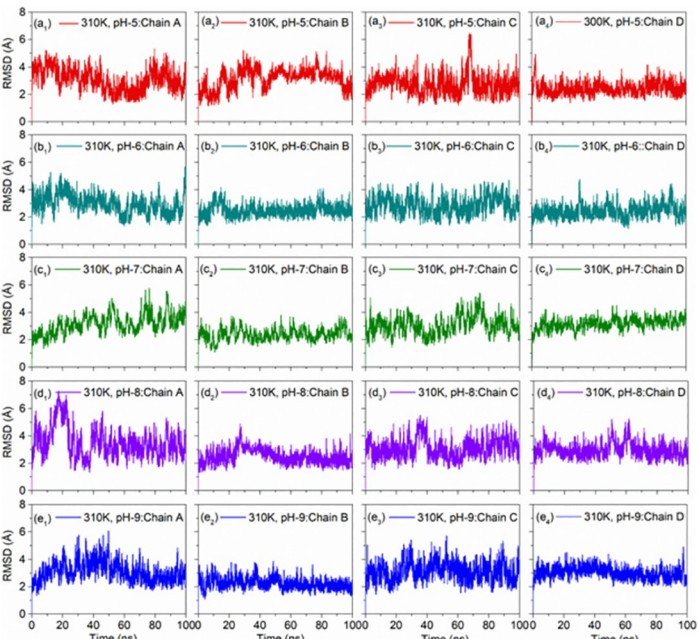

**Fig 8. RMSD plots of the MD simulations of the DNA junction under 310K with altered pH, viz.** ($a_1$-$a_4$) pH5, ($b_1$-$b_4$) pH6, ($c_1$-$c_4$) pH7 as reference, ($d_1$-$d_4$) pH8, and ($e_1$-$e_4$) pH9.

treated under 300K showed occasional deviations higher than 4 Å followed by state of equilibrium, except higher deviations were noted for the Chain A and Chain D at the end of simulation. However, all the chains in DNA junction treated under 310K and pH8 showed considerable stability, except in Chain A ($>$ 6 Å RMSD was noted around 30–40 ns followed by lower RMSD $<$ 5 Å) (Fig 8). Besides, considerable RMSD and stability was noted in all the Chains in DNA junction model treated at 310K with pH9 (Fig 8). These results suggested that 310K and pH9 are ideal parameters for keeping the stable conformation of DNA junction.

**3.2.4. pH dependent RMSF analysis.** The RMSF analysis was performed on the generated DNA junction simulation trajectories to extract the effect of altered pH under two different temperatures (Figs 9 and 10). Notably under both acidic and alkaline environment at 300 and 310K, all the chains showed fluctuations $<$ 5 Å, except higher fluctuations were noted in the Chain C, by comparison to the neutral pH value. Also, terminals of the DNA sequences showed higher RMSF values under all the considered pH environment treated under 300 and 310K. So, from above explanation about RMSF values of all different residues of Chains A, B, C and D and corresponding graphs, DNA junction showed higher local conformational stability under 310K with both acidic and alkaline conditions.

**3.2.5. pH dependent Rg analysis.** To further assess the conformational variations with respect to pH at selected temperatures, Rg for each chain in the DNA model was computed as function of 100 ns dynamics. Notably, chains in the DNA model under both temperatures, i.e., 300K (Fig 11) and 310K (Fig 12) with altered pH concentrations showed steady Rg ($<$ 14 Å) with occasional high peaks ($<$ 20) under acidic and alkaline conditions by comparison to neutral pH.

## 3.3. Essential dynamics of pH dependent trajectories

To get insights into the essential motions of the DNA junction under 300 and 310K temperature with altered pH concentrations, the respective simulation trajectories were computed for

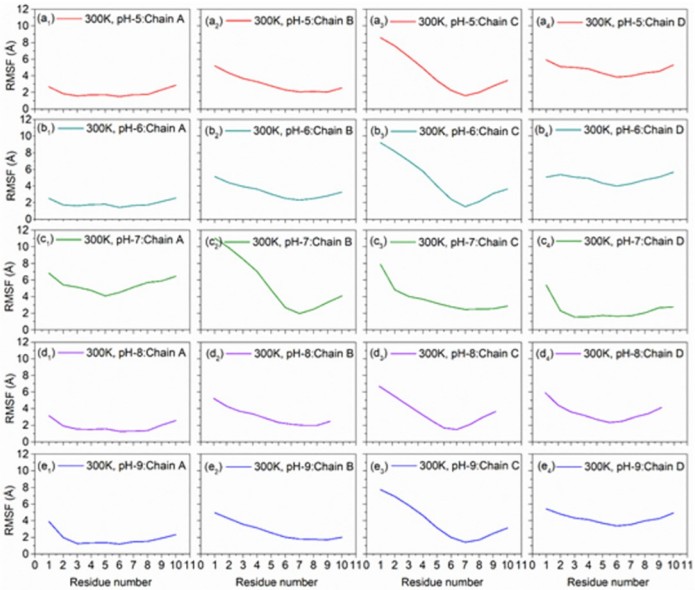

**Fig 9. RMSF plots of the MD simulated DNA Junctions at 300K with altered pH, viz.** (a$_1$-a$_4$) pH5, (b$_1$-b$_4$) pH6, (c$_1$-c$_4$) pH7 as reference, (d$_1$-d$_4$) pH8, and (e$_1$-e$_4$) pH9.

the PCs (Figs 13 and 14). Analysis of scree plot for the DNA junction under 300 and 310K with altered pH showed substantial reduction in essential motions within the first three PCs (Fig 13). Thus, only first three PCs were considered for further analysis.

At 300K, a total of 62.7% (pH5), 75.5% (pH5), 70.1% (pH8), and 71.9% variance (pH9), which was higher than the pH7 (69%) noted at neutral pH. These observations suggested that

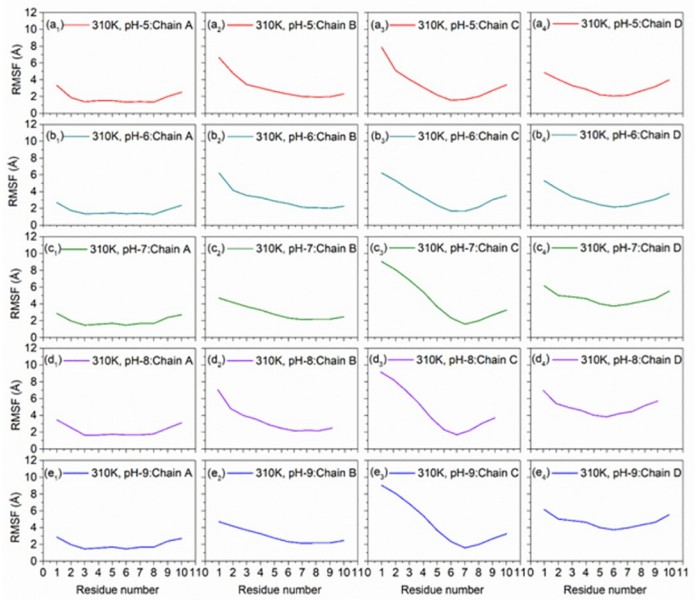

**Fig 10. RMSF plots of the MD simulated DNA Junctions at 310K with altered pH, viz.** (a$_1$-a$_4$) pH5, (b$_1$-b$_4$) pH6, (c$_1$-c$_4$) pH7 as reference, (d$_1$-d$_4$) pH8, and (e$_1$-e$_4$) pH9.

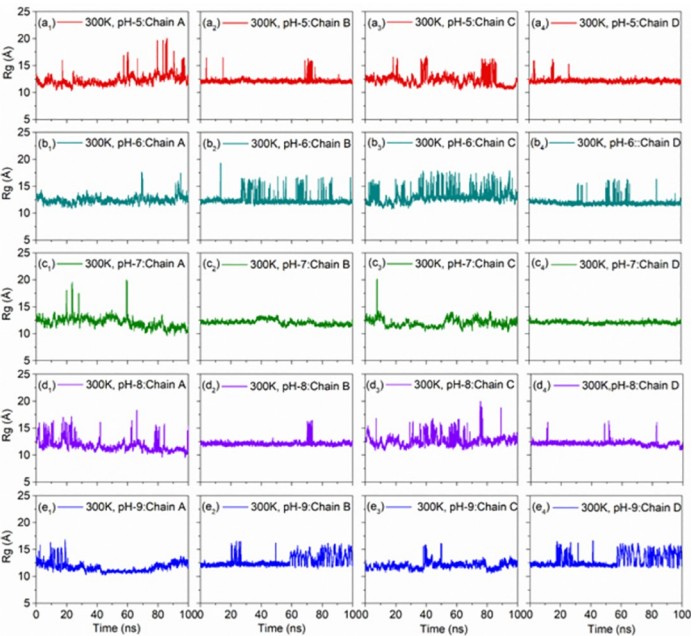

**Fig 11. Variation of radius of gyration (Rg) of all the strands of the DNA Junction with simulation time at 300K with altered pH, viz.** (a$_1$-a$_4$) pH5, (b$_1$-b$_4$) pH6, (c$_1$-c$_4$) pH7 as reference, (d$_1$-d$_4$) pH8, and (e$_1$-e$_4$) pH9.

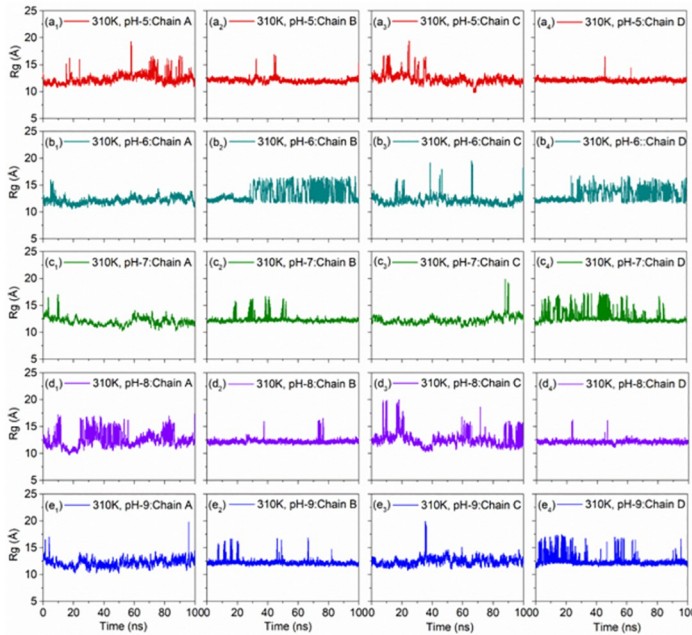

**Fig 12. Variation of radius of gyration (Rg) of all the strands of the DNA Junction with simulation time at 310K with altered pH, viz.** (a$_1$-a$_4$) pH5, (b$_1$-b$_4$) pH6, (c$_1$-c$_4$) pH7 as reference, (d$_1$-d$_4$) pH8, and (e$_1$-e$_4$) pH9.

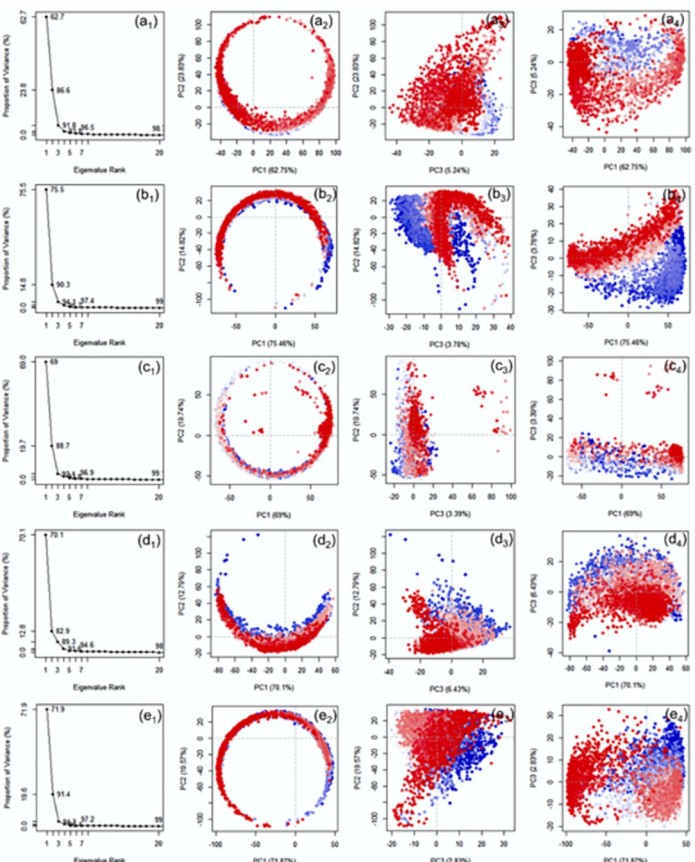

**Fig 13. Principal component analysis for the simulation trajectories generated at 300K with varied pH values, viz.**
pH5 ($a_1$-$a_4$), pH6 ($b_1$-$b_4$), pH7 ($c_1$-$c_4$), pH8 ($d_1$-$d_4$), pH9 ($e_1$-$e_4$). In the X-axis (Eigenvalue Rank) represents the
principal components (PCs) while Y-axis represents the percentage of total variance explained by each individual
principal component. The scatter plots represent the distribution of extracted essential motion between two PCs while
colour change from red to blue through white marks for the periodic fluctuations experienced by the residue during
100 ns MD simulation.

acidic conditions substantially reduced the essential motion by comparison to acidic condi-
tion. These observations were further supported by the dot plot, where no significant differen-
tiation was noted in the clusters plotted for the extracted first three PCs (Fig 13). Similarly, at
310K, a total of 62.7% (pH5), 66.1% (pH6), 66.1% (pH8), and 72.8% (pH9) variance were
noted for the extracted PC1 against 73% variance at neutral pH. These results indicate a reduc-
tion in essential motion under both acidic and alkaline conditions unlike noted fpr the system
under 300K, further supported by distribution of clusters plotted between the three PCs (Fig
14).

## 4. Conclusion

In this paper, we have performed molecular dynamics simulation on the four-way DNA junc-
tion by taking the sequence consisting of Guanine (G) and Cytosine (C) base pairs only. At the
outset, we would like to mention the speciality of this molecule, it consists of ten bases in each
strand and first two bases along the 5′ end was left free as sticky ends, such that it can assemble
with other molecules for periodic arrangements. It means only four base pairs were present in
each arm of the junction molecule. Molecular dynamics for 100.0 ns each, was performed at

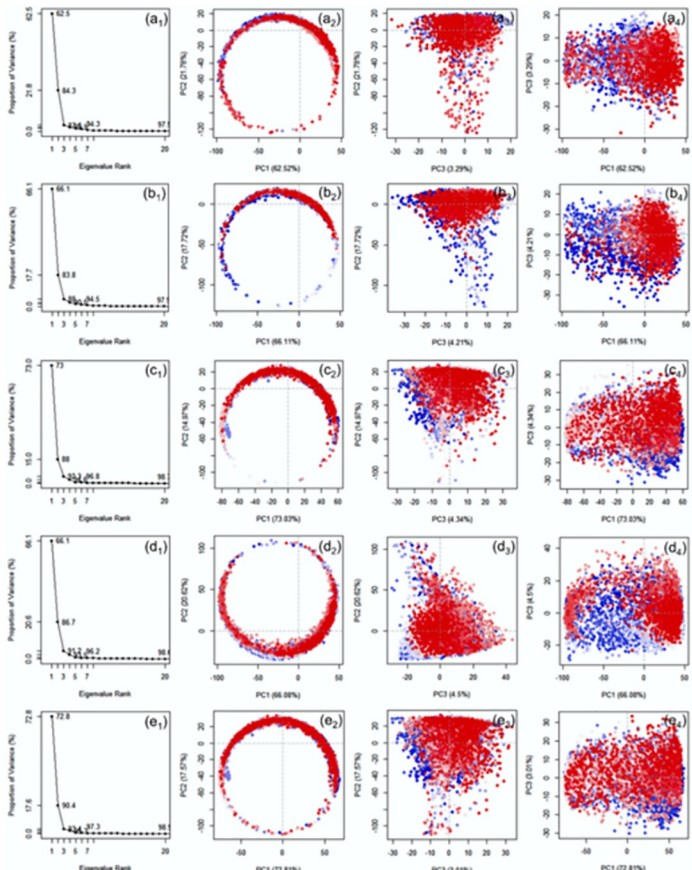

**Fig 14. Principal component analysis for the simulation trajectories generated at 310K with varied pH values, viz.** pH5 ($a_1$-$a_4$), pH6 ($b_1$-$b_4$), pH7 ($c_1$-$c_4$), pH8 ($d_1$-$d_4$), pH9 ($e_1$-$e_4$),. In the X-axis (Eigenvalue Rank) represents the principal components (PCs) while Y-axis represents the percentage of total variance explained by each individual principal component. The scatter plots represent the distribution of extracted essential motion between two PCs while colour change from red to blue through white marks for the periodic fluctuations experienced by the residue during 100 ns MD simulation.

pH values 5, 6, 7, 8 and 9, at three different temperatures 200K, 300K and 310K respectively. During the long duration of dynamical pathway, the variation in structure and conformation was investigated by taking snap shots at the interval of 10 ns. Prima facie it seems that the structure of the junction molecule stay put during dynamics simulation, but a close observation reveals variation in most of the structural and geometrical parameters i.e., RMSD, RMSF, radius of gyration, torsion angle parameters, helix axis parameters etc. as explained in results and discussion section. The conformational and torsion angle parameters for DNA junction give a clear picture of dynamical behaviour of the junction molecule. The changes in the parameters are albeit small but indicate that at the junction cross over, there occurs a hinge point where the junction seems to be folding. The conformational analysis of the d (CGGCGGCCGC)$_4$ four-way Holliday junction at 200K suggested that DNA junction maintained the conformation, i.e., B type DNA, and structural integrity during 100 ns MD simulation, supported by statistical analysis of the trajectory. However, on rising the temperature of simulation to 300K and 310K, there are abrupt changes in the values of the parameters which gives indication of the fact that the DNA junction experienced both structural and conformational changes during MD simulation followed by a state of equilibrium with altered

parameters by comparison to crystal structure. Also, when pH values were shifted from neutral to acidic, it was observed that at certain concentration of acidic solution, the structural variations were identical to those obtained at neutral pH. Similar was the case for changing the pH values from neutral to alkaline. It means the alteration of pH from acidic to alkaline depends on the protonation of ions present in the solution. Albeit acidic ash is harmful and vulnerable to illness while alkaline is protective to health. Finally, we may say that, unlike enzymes and antibodies, DNA junction periodic arrangements can be readily synthesized and regenerated for multiple uses, since they are temperature sensitive and pH dependent, and can be used for temperature and pH dependent DNA-based nanotechnology applications.

## Supporting information

**S1 Table. Torsion angles parameters of the initial structure of the DNA junction, final structure after 100 ns MD simulation done at 200K temperature, final structure after 100 ns MD simulation done at 300K temperature, final structure after 100 ns MD simulation done at 310K temperature.**
(PDF)

**S2 Table. Helix axis parameters(intra) of the initial structure of the DNA Junction, final structure after 100 ns MD simulation done at 200K temperature, final structure after 100 ns MD simulation done at 300K temperature, final structure after 100 ns MD simulation done at 310K temperature.**
(PDF)

**S3 Table. Torsion angles parameters of the MD simulated structures at 300K and pH values 5, 6, 7, 8 and 9.**
(PDF)

**S4 Table. Torsion angles parameters of the MD simulated structures at 310K and pH values 5, 6, 7, 8 and 9.**
(PDF)

**S5 Table. Local base-pair, local base-pair step and local base-pair helical parameters of the MD simulated structures at 300K and pH values 5, 6, 7, 8 and 9.**
(PDF)

**S6 Table. Local base-pair, local base-pair step and local base-pair helical parameters of the MD simulated structures at 310K and pH values 5, 6, 7, 8 and 9.**
(PDF)

## Author Contributions

**Conceptualization:** Akanksha Singh, Shiv Bharadwaj, Umesh Yadava.

**Data curation:** Akanksha Singh, Shiv Bharadwaj, Umesh Yadava.

**Formal analysis:** Akanksha Singh.

**Funding acquisition:** Hesham Hasssan, Rashmi Rana.

**Investigation:** Akanksha Singh, Shiv Bharadwaj, Umesh Yadava.

**Methodology:** Akanksha Singh, Shiv Bharadwaj, Umesh Yadava.

**Resources:** Nitin Kumar Kamboj.

**Supervision:** Shiv Bharadwaj, Rashmi Rana, Umesh Yadava.

**Validation:** Ramesh Kumar Yadav, Nitin Kumar Kamboj, Shiv Bharadwaj, Umesh Yadava.

**Visualization:** Shiv Bharadwaj.

**Writing – original draft:** Akanksha Singh.

**Writing – review & editing:** Ramesh Kumar Yadav, Ali Shati, Hesham Hasssan, Shiv Bharadwaj, Rashmi Rana, Umesh Yadava.

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
