## [Decision Letter · Decision Letter 0]

17 Aug 2022

PONE-D-22-19429Understanding the self-assembly dynamics of A/T absent ‘four-way DNA junctions with sticky ends’ at altered physiological conditions through molecular dynamics simulationsPLOS ONE Dear Dr. Yadava,

Thank you for submitting your manuscript to PLOS ONE. After careful consideration, we feel that it has merit but does not fully meet PLOS ONE’s publication criteria as it currently stands. Therefore, we invite you to submit a revised version of the manuscript that addresses the points raised during the review process.

We look forward to receiving your revised manuscript.

Kind regards,

Ki Hyun Nam

Academic Editor

PLOS ONE

Journal Requirements:

Reviewers' comments:

Reviewer's Responses to Questions

**Comments to the Author**

1. Is the manuscript technically sound, and do the data support the conclusions?

Reviewer #1: Partly

Reviewer #2: Partly

2. Has the statistical analysis been performed appropriately and rigorously? 

Reviewer #1: I Don't Know

Reviewer #2: Yes

3. Have the authors made all data underlying the findings in their manuscript fully available?

Reviewer #1: Yes

Reviewer #2: Yes

4. Is the manuscript presented in an intelligible fashion and written in standard English?

Reviewer #1: Yes

Reviewer #2: Yes

5. Review Comments to the Author

Reviewer #1: The manuscript by Singh and coworkers reports on the self-assembly of DNA junctions modeled using molecular dynamics simulations. The Authors selected the Holliday junction comprising four two-fold helical arms with sticky ends. Specifically, the DNA sequence d(CGGCGGCCGC)4 was used to determine the pathways and outcomes of the self-assembly under physiological conditions. To that end, extensive MD calculations were performed in which the effect of temperature and pH on the conformation of the junction was examined. The main result of these investigations was a bunch of snapshots and a large set of structural parameters including fluctuations in coordinates and torsion angles. The methodology of the calculations was based on a ready-to-use MD software and it was described in section 2. Although the study provides a lot of data it looks much like a technical report. I do not see the clue of the theoretical studies described in the manuscript. In particular, sufficient physical explanation of the simulated effects is missing. What are the reasons for the observed structural changes in the conformation of the Junction when external parameters are changed? The conclusions of the paper should be refined and main new physical effects predicted by calculations should be clearly summarized. In conclusion, the paper can be accepted after revision which included the following points.

1) The Authors claim that their study refers to altered physiological conditions. However, some of the values of pH and temperature can hardly be called altered physiological (allowing the normal functioning of living organisms). While the temperature 300 K and pH close 7 are OK it is hard to consider the remaining values physiological. This refers especially to the unnaturally low values of temperature 100 K and 200K. What is the meaning of using such low temperature for biochemical processes running in living organisms?

2) What is the physical origin of the minima observed in the plot of RMSF shown for chainB and ChainA at 300K? (Figure 5) The same refers to Figure S2 (c-type plots).

3) The tables comprising large number of computed values should be moved to the Supporting Information Section (Tables 1 and 2). Instead of these tables , Figure S1 ,or better, pH effect at 300 K should be included and discussed in the main text.

4) A the beginning of section 3.6 the Authors write “It has been observed from the above discussion that the junction with sequence d(CGGCGGCCGC)4 exhibit better stability at lower temperatures” and a few lines before “On further raising the temperature to 300K there is abrupt change in the values of

parameters, it shows that the junction gets completely unstable. So, we conclude that the junction gets

unstable with rise in temperature.” Are these conclusions anything new? They seem to be quite obvious: when temperature rises kinetic energy of atoms increases leading to enhanced motion and breakup of structures they formed. This should be explained.

5) Also, in the conclusion section one can read” When pH values were shifted from neutral to acidic, it was observed that at certain concentration of acidic solution, the structural variations were identical to those obtained at neutral pH.” and later “Finally, we may say that, unlike enzymes and antibodies, DNA junction periodic arrangements can be readily synthesized and regenerated for multiple uses, since they are temperature sensitive and pH dependent.” From figure S1 it follows that changes in the structure of the Junction are observable mostly at pH equal to 7. What is the reason for this phenomenon? Is this sensitivity limited just to pH range close to 7? If so, more detailed studies at pH values close to 7 would be more informative and physiologically relevant.

6) The number of water molecules and ions used in the simulations should be explicitly stated.

7) At which temperature the data from Table 3 were obtained?

Reviewer #2: This could be an interesting manuscript as it deals with an interesting concept. However, it has major issues that prevent it from being considered, whose reasons are as follows:

1. In Figure 3 authors report a large disruption at 90 ns at 300 K, which is clearly visible, with the snapshot at 100 ns being again "normal". However, authors did not try to explain or discuss it. Was this a random event or was it systematically observed? Further discussion on this must be done.

2. In Figure 4, for each temperature all graphs report to be following chain A, but when reading the manuscript one finds that these should rather be for chains A-D. Please verify and correct.

3. Still concerning data presented in Figure 4, specifically for 300 K, why is the data in Figure 4c4showing a negative slope with increasing dynamics? This seems opposite to what is being discussed in the text as leading to disruption. Furthermore, in the text just above Figure 4, authors state that "we conclude that DNA junction is temperature sensitive", which, although supported by the data is a known phenomenon as elevated temperatures disrupt intermolecular interactions and therefore its importance as a conclusion should be minimized. Again, authors should provide an explanation for the negative slope of the chart in Figure 4c4.

4. The most interesting feature of the manuscript, which the pH simulation and how the DNA complex responds to the different pH environment was relegated to the SI material. In addition, authors simulated it for the 100 K temperature, which is meaningless in a real purpose application, being contrary to what they state in the Introduction, as these structures may be used as potential drug delivery structures. I would expect authors to test the pH response at 300 K as well as this would be more realistic to a human body's regular temperature.

The bottom line of this manuscript is that it focused on temperature effects in the discussion, being redundant in the conclusions, while leaving out the most interesting part - pH simulation, while leaving not testing the simulation at 300 K. As such I cannot recommend it for publication.

6. PLOS authors have the option to publish the peer review history of their article (what does this mean?). If published, this will include your full peer review and any attached files.

Reviewer #1: No

Reviewer #2: No

---

## [Author Response · Author response to Decision Letter 0]

23 Oct 2022

#Review Comments to the Author

Reviewer #1: The manuscript by Singh and co-workers reports on the self-assembly of DNA junctions modeled using molecular dynamics simulations. The Authors selected the Holliday junction comprising four two-fold helical arms with sticky ends. Specifically, the DNA sequence d(CGGCGGCCGC)4 was used to determine the pathways and outcomes of the self-assembly under physiological conditions. To that end, extensive MD calculations were performed in which the effect of temperature and pH on the conformation of the junction was examined. The main result of these investigations was a bunch of snapshots and a large set of structural parameters including fluctuations in coordinates and torsion angles. The methodology of the calculations was based on ready-to-use MD software and it was described in section 2. 

#Comment a: Although the study provides a lot of data it looks much like a technical report. I do not see the clue of the theoretical studies described in the manuscript. In particular, sufficient physical explanation of the simulated effects is missing. 

Response: Thank you for your constructive remarks. We have added and discussed the base composition with respect to altered temperature and pH. Also, we have attempted to add the effect of entropy and protonation under the discussion section in revised manucript for your kind consideration. Thank you!

#Comment b: What are the reasons for the observed structural changes in the conformation of the Junction when external parameters are changed? 

Response: Thank you, we understand it could happen due to change in charge due to protonation or electrostatic charge development due to increase in entropy of the system. A suitable discussion has been added in the revised manucript for your consideration. 

#Comment c: The conclusions of the paper should be refined and main new physical effects predicted by calculations should be clearly summarized. In conclusion, the paper can be accepted after revision which included the following points.

Response: We agreed with your given suggestion. As per your given suggestions, the rectifications have been added in the revised manucript for your kind consideration.

#Comment 1: The Authors claim that their study refers to altered physiological conditions. However, some of the values of pH and temperature can hardly be called altered physiological (allowing the normal functioning of living organisms). While the temperature 300 K and pH close 7 are OK it is hard to consider the remaining values physiological. This refers especially to the unnaturally low values of temperature 100 K and 200K. What is the meaning of using such low temperature for biochemical processes running in living organisms?

Response: We agreed with your given remarks. We attempted to study the stick end DNA junction with respect to alter physiological conditions, including those allowed normal functioning of living organisms, with an aim for DNA-based nanotechnology applications. Besides, we agreed with your given suggestions, thus, dataset for the 100K has been removed in the revised version. Also, we consider the evaluation of selected DNA HJ junction at 310 K (which is equilivent to human body temperature) under altered pH concentrations and added has been in the revised manucript for your consideration. Thank you!

#Comment 2: What is the physical origin of the minima observed in the plot of RMSF shown for chainB and ChainA at 300K? (Figure 5) The same refers to Figure S2 (c-type plots).

Response: As the temperature increases, fluctuations of the residues increase. The minimum fluctuations are observed where binding is tight as compared to other residues. This discussion is now included within the manuscript.

#Comment 3: The tables comprising large number of computed values should be moved to the Supporting Information Section (Tables 1 and 2). Instead of these tables, Figure S1, or better, pH effect at 300 K should be included and discussed in the main text.

Response: Thanks to Reviewer for the suggestion, we have now shifted tables 1 & 2 to the Supplementary information section and RMSD plots of the MD simulations of the DNA junction at 310 K with different pH, i.e. (a) pH5, (b) pH6, (c) pH8, and (d) pH9 is now included.

#Comment 4: A the beginning of section 3.6 the Authors write “It has been observed from the above discussion that the junction with sequence d(CGGCGGCCGC)4 exhibit better stability at lower temperatures” and a few lines before “On further raising the temperature to 300K there is abrupt change in the values of parameters, it shows that the junction gets completely unstable. So, we conclude that the junction gets unstable with rise in temperature.” Are these conclusions anything new? They seem to be quite obvious: when temperature rises kinetic energy of atoms increases leading to enhanced motion and breakup of structures they formed. This should be explained.

Response: Thanks for the suggestion, we have included the line “this is because of the fact that kinetic energy of atoms increases with the rise of temperature that leads to the enhanced vibrations and instability which finally results in to the disruption of the DNA junction” at the place of “So, we conclude that the junction gets unstable with rise in temperature”.

#Comment 5: Also, in the conclusion section one can read” When pH values were shifted from neutral to acidic, it was observed that at certain concentration of acidic solution, the structural variations were identical to those obtained at neutral pH.” and later “Finally, we may say that, unlike enzymes and antibodies, DNA junction periodic arrangements can be readily synthesized and regenerated for multiple uses, since they are temperature sensitive and pH dependent.” From figure S1 it follows that changes in the structure of the Junction are observable mostly at pH equal to 7. What is the reason for this phenomenon? Is this sensitivity limited just to pH range close to 7? If so, more detailed studies at pH values close to 7 would be more informative and physiologically relevant.

Response: This study has been carried out to verify the fact that DNA junctions are temperature sensitive and pH dependent which play crucial role in the DNA tweezers, DNA nanotechnology, DNA biomarkers and DNA biosensor.

#Comment 6: The number of water molecules and ions used in the simulations should be explicitly stated.

Response: The numbers of water molecules and ions used in the simulations have now been stated in the methodology section.

#Comment 7: At which temperature the data from Table 3 were obtained?

Response: since DNA junction has dynamical structures at larger temperatures, its groove width will be more precise at lower temperature. Hence, the variations of groove widths in table 3(now table 1 in revised manuscript) are discussed at 100K temperature. It is now mentioned within the manuscript and title of the table.

Reviewer #2: This could be an interesting manuscript as it deals with an interesting concept. However, it has major issues that prevent it from being considered, whose reasons are as follows:

#Comment 1: In Figure 3 authors report a large disruption at 90 ns at 300 K, which is clearly visible, with the snapshot at 100 ns being again "normal". However, authors did not try to explain or discuss it. Was this a random event or was it systematically observed? Further discussion on this must be done.

Response: It has been shown that at 300K the structure is unstable. This is the reason why a disruption is visible in one of the snapshots at 300K in Figure 3. 

#Comment 2: In Figure 4, for each temperature all graphs report to be following chain A, but when reading the manuscript one finds that these should rather be for chains A-D. Please verify and correct.

Response: Thank you for pointing out. It is typing mistake, which has now been corrected. 

#Comment 3: Still concerning data presented in Figure 4, specifically for 300 K, why is the data in Figure 4c4 showing a negative slope with increasing dynamics? This seems opposite to what is being discussed in the text as leading to disruption. Furthermore, in the text just above Figure 4, authors state that "we conclude that DNA junction is temperature sensitive", which, although supported by the data is a known phenomenon as elevated temperatures disrupt intermolecular interactions and therefore its importance as a conclusion should be minimized. Again, authors should provide an explanation for the negative slope of the chart in Figure 4c4.

Response: The figure 4 shows RMSDs of the chains A, B, C and D respectively at (a) 100K, (b) 200K and (c) 300K temperature. The Figure 4c indicate that the RMS deviations of the chains A, B, C increases with simulation time while that of chain D slightly decreases. This is natural that three chains are detaching from fourth one (Chain D), hence the RMS deviations are so observed. 

#Comment 4: The most interesting feature of the manuscript, which the pH simulation and how the DNA complex responds to the different pH environment was relegated to the SI material. In addition, authors simulated it for the 100 K temperature, which is meaningless in a real purpose application, being contrary to what they state in the Introduction, as these structures may be used as potential drug delivery structures. I would expect authors to test the pH response at 300 K as well as this would be more realistic to a human body's regular temperature.

Response: Thanks to the reviewer for the suggestion. We have now performed MD simulation and conformational analysis of the DNA junction at 300 K and 310 K temperature. 

#Comment 5: The bottom line of this manuscript is that it focused on temperature effects in the discussion, being redundant in the conclusions, while leaving out the most interesting part - pH simulation, while leaving not testing the simulation at 300 K. As such I cannot recommend it for publication.

Response: The conclusion has been revised as per reviewers’ suggestions.

---

## [Decision Letter · Decision Letter 1]

15 Nov 2022

PONE-D-22-19429R1Understanding the self-assembly dynamics of A/T absent ‘four-way DNA junctions with sticky ends’ at altered physiological condition through molecular dynamics simulationsPLOS ONE

Dear Dr. Yadava,

Thank you for submitting your manuscript to PLOS ONE. After careful consideration, we feel that it has merit but does not fully meet PLOS ONE’s publication criteria as it currently stands. Therefore, we invite you to submit a revised version of the manuscript that addresses the points raised during the review process. Please submit your revised manuscript by Dec 30 2022 11:59PM. If you will need more time than this to complete your revisions, please reply to this message or contact the journal office at plosone@plos.org. Please include the following items when submitting your revised manuscript:A rebuttal letter that responds to each point raised by the academic editor and reviewer(s). You should upload this letter as a separate file labeled 'Response to Reviewers'.A marked-up copy of your manuscript that highlights changes made to the original version. You should upload this as a separate file labeled 'Revised Manuscript with Track Changes'.An unmarked version of your revised paper without tracked changes. You should upload this as a separate file labeled 'Manuscript'.If applicable, we recommend that you deposit your laboratory protocols in protocols.io to enhance the reproducibility of your results. Protocols.io assigns your protocol its own identifier (DOI) so that it can be cited independently in the future. For instructions see: https://journals.plos.org/plosone/s/submission-guidelines#loc-laboratory-protocols. Additionally, PLOS ONE offers an option for publishing peer-reviewed Lab Protocol articles, which describe protocols hosted on protocols.io. Read more information on sharing protocols at https://plos.org/protocols?utm_medium=editorial-email&utm_source=authorletters&utm_campaign=protocols.

We look forward to receiving your revised manuscript.

Kind regards,

Ki Hyun Nam

Academic Editor

PLOS ONE

Journal Requirements:

**Comments to the Author**

Reviewer #1: The Authors addressed all of my comments in detail. I am OK with these explanations and changes made to manuscript. It can be now published in PLOS ONE

Reviewer #2: The manuscript in its revised version has improved massively as in my view all concerns raised by reviewers were addressed. From that point of view the manuscript may be accepted for publication.

However, I found some issues with the captions of Figures 5, 13 and 14 as the description of some of the images is missing or misleading. Authors must revise this.

---

## [Author Response · Author response to Decision Letter 1]

22 Nov 2022

#Journal Requirements:

Response: Thank you very much for the suggestion. The reference list have been reviewed and corrected. As per suggestion, some references have been replaced by other relevant current references. Changes have been highlighted in the revised manuscript with track changes. Retracted references have not been included in the manuscript.

# Review Comments to the Author

Reviewer #1: The Authors addressed all of my comments in detail. I am OK with these explanations and changes made to manuscript. It can be now published in PLOS ONE.

Response: Thanks to the Reviewer for his commend.

Reviewer #2: The manuscript in its revised version has improved massively as in my view all concerns raised by reviewers were addressed. From that point of view the manuscript may be accepted for publication.

Response: Thanks to the Reviewer for his commend.

Comment: However, I found some issues with the captions of Figures 5, 13 and 14 as the description of some of the images is missing or misleading. Authors must revise this.

Response: Thank you for pointing out. It is typing mistake, which have now been corrected in the re-revised manuscript.

---

## [Editor Report · Decision Letter 2]

23 Nov 2022

Understanding the self-assembly dynamics of A/T absent ‘four-way DNA junctions with sticky ends’ at altered physiological condition through molecular dynamics simulations

PONE-D-22-19429R2

Dear Dr. Yadava,

We’re pleased to inform you that your manuscript has been judged scientifically suitable for publication and will be formally accepted for publication once it meets all outstanding technical requirements.

Kind regards,

Ki Hyun Nam

Academic Editor

PLOS ONE

---

## [Editor Report · Acceptance letter]

5 Dec 2022

PONE-D-22-19429R2 

Understanding the self-assembly dynamics of A/T absent ‘four-way DNA junctions with sticky ends’ at altered physiological conditions through molecular dynamics simulations 

Dear Dr. Yadava:

I'm pleased to inform you that your manuscript has been deemed suitable for publication in PLOS ONE. Congratulations! Your manuscript is now with our production department. 

Kind regards, 

on behalf of

Dr. Ki Hyun Nam 

Academic Editor

PLOS ONE